# Plant Cell-Engineered Gold Nanoparticles Conjugated to Quercetin Inhibit SARS-CoV-2 and HSV-1 Entry

**DOI:** 10.3390/ijms241914792

**Published:** 2023-09-30

**Authors:** James Elste, Sangeeta Kumari, Nilesh Sharma, Erendira Palomino Razo, Eisa Azhar, Feng Gao, Maria Cuevas Nunez, Wasim Anwar, John C. Mitchell, Vaibhav Tiwari, Shivendra Sahi

**Affiliations:** 1Department of Microbiology & Immunology, College of Graduate Studies, Midwestern University, Downers Grove, IL 60515, USA; jelste@midwestern.edu (J.E.); eisa.azhar@midwestern.edu (E.A.); 2Department of Biology, Saint Joseph’s University, University City Campus, Philadelphia, PA 19131, USA; sangeeta.mscbhu@gmail.com (S.K.); wanwar@sju.edu (W.A.); 3Department of Biology, Western Kentucky University, Bowling Green, KY 42101, USA; nilesh.sharma@wku.edu; 4College of Dental Medicine, Midwestern University, Downers Grove, IL 60515, USA; erendira.palominoraz@midwestern.edu (E.P.R.); fgao@midwestern.edu (F.G.); caritocuevas@gmail.com (M.C.N.); jmitch@midwestern.edu (J.C.M.)

**Keywords:** viral entry, virus host cell interactions, nanoparticles

## Abstract

Recent studies have revealed considerable promise in the antiviral properties of metal nanomaterials, specifically when biologically prepared. This study demonstrates for the first time the antiviral roles of the plant cell-engineered gold nanoparticles (pAuNPs) alone and when conjugated with quercetin (pAuNPsQ). We show here that the quercetin conjugated nanoparticles (pAuNPsQ) preferentially inhibit the cell entry of two medically important viruses—severe acute respiratory syndrome coronavirus 2 (SARS-CoV-2) and herpes simplex virus type-1 (HSV-1) using different mechanisms. Interestingly, in the case of SARS-CoV-2, the pre-treatment of target cells with pAuNPsQ inhibited the viral entry, but the pre-treatment of the virus with pAuNPsQ did not affect viral entry into the host cell. In contrast, pAuNPsQ demonstrated effective blocking capabilities against HSV-1 entry, either during the pre-treatment of target cells or by inducing virus neutralization. In addition, pAuNPsQ also significantly affected HSV-1 replication, evidenced by the plaque-counting assay. In this study, we also tested the chemically synthesized gold nanoparticles (cAuNPs) of identical size and shape and observed comparable effects. The versatility of plant cell-based nanomaterial fabrication and its modification with bioactive compounds opens a new frontier in therapeutics, specifically in designing novel antiviral formulations.

## 1. Introduction

Interactions between biomolecules and nanoparticles (NPs) offer their use in various medical interventions, including developing next-generation viral entry inhibitors [1,2,3,4]. Such interactions are often multivalent and involve multiple copies of receptors and ligands that bind in a coordinated manner, resulting in drastically enhanced specificities [5]. The attachment and entry of viruses into the host cell is an outcome of such multivalent interactions between viral envelope glycoprotein and cell membrane receptor [6]. For example, in the case of severe acute respiratory syndrome coronavirus 2 (SARS-CoV-2), the virus spike (S) glycoprotein interacts with angiotensin-converting enzyme-2 (ACE-2) and multiple other cell surface receptors, including cell surface heparan sulfate to facilitate viral entry [7,8,9,10,11]. Similarly, the actions of numerous herpes simplex virus type-1 (HSV-1) glycoproteins are involved in promoting virus entry by using multiple cell surface receptors [12,13,14]. Interfering with these recognition events between the virus and the host cell is one of the most promising strategies for developing novel pre-and post-exposure prophylaxis, microbicides, and therapeutics to prevent viral infections [15].

In this study, we were interested in testing the effect of plant cell-engineered gold nanoparticles (pAuNPs) in the presence or absence of conjugated quercetin on SARS-CoV-2 and HSV-1 viruses as they pose a significant risk, specifically in dental clinic settings and procedures with no commercial antiviral drugs available to target the entry of pathogenic variants of SARS-CoV-2 or with the emerging drug-resistant strains of HSV-1 [16,17,18]. Interestingly, our group reported the development of plant-engineered gold nanoparticles (pAuNPs) [19] conjugated with the phytochemical quercetin [personal communication]. A range of bioactive secondary metabolites of plant origin, including polyphenols and flavonoids, is touted for remarkable antimicrobial, antioxidative, and anti-inflammatory efficacies [20,21,22]. This concept inspired us to investigate if flavonoid-conjugated specific AuNPs could generate antiviral effects against SARS-CoV-2 and HSV-1. In parallel, we tested the synthetic counterpart of AuNPs to compare its antiviral roles to those of pAuNPs. Using cell culture-based assays with the reporter viruses, we demonstrate here that pAuNPs conjugated with quercetin––at the non-toxic concentration––effectively block SARS-CoV-2 and HSV-1 entry into host cells.

Interestingly, our results show mechanistic differences in the modes of action of pAuNPs in blocking two different viruses, as they exhibit pleiotropicity, i.e., the ability to bind differentially to multiple targets in different viruses. Since both viruses rely on numerous cell surface receptors to access different host cell types, attenuating or inhibiting these pathways with unique pleiotropic inhibitors is possible. A uniquely designed nanomaterial in the cellular environment such as pAuNPs when conjugated with a suitable phytochemical including quercetin (pAuNPsQ) can affect desired viral entry pathways. Given that many patients visiting dental clinics may be immunologically compromised or unvaccinated, these patients are more likely to be vulnerable to infection and, if infected, may present a higher viral load, increasing risk to dental workers [23,24]. In addition, many devices used in dental practices, such as ultrasonic scalers, air-water syringes, and dental handpieces, are known to generate water aerosols. Therefore, the above practices increase the risk of exposure to infectious respiratory pathogens, including SARS-CoV-2, to patients and healthcare workers [25,26]. Hence, the strategies directed to minimize the risk of SARS-CoV-2 transmission by targeting the ability of viral particles to fuse with the host cells or by reducing the titers of SARS-CoV-2 in the saliva of infected patients seems to be a promising approach. This is particularly important in high-risk procedures in dental care. Against this backdrop, developing novel practices that either dismantle or prevent the virus from infecting a new host provide valuable therapeutic interventions against such respiratory tract viruses [27,28,29]. Our use of a novel edible nanomaterial conjugated with the natural flavonoid (quercetin) in the present study establishes its potential for future antiviral applications, particularly in tooth cavity fillings where an amalgam containing mercury (Hg), silver (Ag), tin (Sn), and other metals is used [30]. Further refinement of the cell-based nanoparticle fabrication system using a bioreactor will likely advance plant-based antiviral therapeutics in the future.

## 2. Results

Characterization of plant cell-engineered gold nanoparticles. Harvested and purified plant cell-engineered gold nanoparticles shown in materials and methods (Figure 1) conjugated with quercetin (pAuNPsQ) and or without quercetin (pAuNPs) were characterized as follows.

UV-Visible spectroscopy: As indicated in Figure 2A, the UV-Visible absorbance spectrum of the particles eluted from the gel permeation column showed a peak around 550 nm, pointing to the nature of gold nanoparticles. However, the quercetin-coated particles (pAuNPsQ) exhibited a peak around 565 nm (Figure 2B). The shift in peak to higher wavelengths indicates the coating of quercetin on the surface of nanoparticles.

### 2.1. Size & Zeta (ζ) Potential

Particle size and zeta potential of the nanoparticles were determined by the Dynamic light scattering (DLS) method. DLS is concerned with measurement of particles suspended within a liquid. DLS analysis revealed that the plant-derived gold nanomaterial (pAuNPs) had an average particle size of 21.04 nm (Figure 2C), which increased to 43.82 nm after quercetin coating in pAuNPsQ (Figure 2D). Similar effects were observed in Zeta potential values: pAuNPs exhibited a **ζ** potential of −23.5 mV (Figure 3A), which dropped further to −28.6 mV in pAuNPsQ (Figure 3B).

### 2.2. Fourier Transform Infra-Red Spectroscopy (FTRI)

Analysis of the Fourier-transform infrared spectroscopy (FTIR) generated from pAuNPs points to characteristic peaks at 2896 cm^−1^ and 2977 cm^−1^, which correspond to amino (-N-H) and hydroxyl (-O-H) functional groups present on the surface of these nanoparticles (Figure 3C). This spectrum also includes another peak corresponding to alkyl groups present on the surface of pAuNP. The quercetin-coated nanomaterial (pAuNPsQ) spectrum exhibits the peaks mainly due to hydroxyl (-O-H), aromatic alkene, and alkyl functional groups (Figure 3D). The peak representing hydroxyl group in pAuNPsQ can be noted at 2972 cm^−1^, which indicates a shift from its original position (2977 cm^−1^) in pAuNPs.

### 2.3. Quantification of Cytotoxicity Associated with the Nanoparticles Using LDH Assay

The potential cytotoxic side effect of the plant cell-derived (pAuNPs) and chemically synthesized (cAuNPs) gold nanoparticles with and without quercetin on the target cells was measured by the release of LDH—an enzyme that is an indicator of cellular toxicity [31]. In this experiment, pAuNPs and cAuNPs were pre-treated separately on HEK293T-ACE2 (Figure 4A) and HeLa cells (Figure 4B) using 10 µg/mL and 20 µg/mL dosages. As evident in Figure 4, the tested concentration of pAuNPs and cAuNPs showed minimal toxicity compared to the spontaneous release negative control and significantly less toxicity than the maximum LDH release represented by the positive control. Taken together, the concentrations of the pAuNPs and cAuNPs used in this study were non-toxic to cells.


**Preincubation of the target cells with quercetin-conjugated plant-derived gold nanoparticles (pAuNPsQ) and or chemically synthesized quercetin-conjugated gold nanoparticles (cAuNPsQ) inhibited SARS-CoV-2 entry in HEK293T-ACE2 cells.**


The effect of unconjugated (pAuNPs and cAuNPs) and quercetin-conjugated (pAuNPsQ and cAuNPsQ) nanoparticles of both origins were tested against SARS-CoV-2 cell entry using HEK293T-ACE-2 cells, which overexpress the human ACE2 receptor [7]. Two approaches were used in this experiment. In the first approach, the target cells were pre-treated with the nanoparticles mentioned above in a dosage-dependent manner for 1 h at 37 °C before challenging the luciferase base reporter pseudovirus SARS-CoV-2. Forty-eight hours post-infection, viral entry was determined by measuring luciferase activity as described in Section 4. In parallel, the HEK293T-ACE2 cells transduced with the pseudovirus- SARS-CoV-2 in the absence of nanoparticles were used as a positive control, while the mock (PBS) transduced HEK293T-ACE2 cells served as a negative control. As shown in Figure 5A, except for unconjugated cAuNPs, all tested nanoparticles inhibited SARS-CoV-2 cell entry. Specifically, nanoparticles conjugated to quercetin (pAuNPsQ and cAuNPsQ) showed a significant dosage-dependent effect on SARS-CoV-2 entry. Since the antiviral effect of the above nanoparticles was evident using pseudovirus viral entry assay, we rationalized testing if nanoparticles could impair the virus. Therefore, in the second approach, we tested the impact of both quercetin-conjugated and unconjugated nanoparticles on virus neutralization. In this experiment, the pseudovirus SARS-CoV-2 was pre-treated with the nanoparticles individually for one hour before transducing the target HEK293T-ACE-2 cells. Mock-treated SARS-CoV-2 transduced cells were used as the positive control, while uninfected or untransduced cells were used as the negative control. As indicated in Figure 5B, the blocking effect of pAuNPsQ and cAuNPsQ were lost during the virus neutralization assay suggesting that the effective nanoparticles likely act on cell surface receptors.


**Both pre-treatment and virus neutralization assay with quercetin-conjugated plant-derived gold nanoparticle (pAuNPsQ) and or quercetin-conjugated chemically synthesized gold nanoparticles (cAuNPsQ) efficiently blocks HSV-1 entry.**


We next verified if the anti-SARS-CoV-2 activity of the pAuNPsQ and cAuNPsQ can also impact HSV-1, a medically significant virus commonly reported in dental practice [32]. We first pre-treated target HeLa cells individually with both quercetins conjugated or unconjugated nanoparticles for 1 h before challenging with the β-galactosidase reporter HSV-1 virus. The mock-treated and mock-infected cells were considered positive and negative controls, respectively. As shown in Figure 6A, only quercetin-conjugated nanoparticles (pAuNPsQ and cAuNPsQ) inhibited the HSV-1 cell entry in a dosage-dependent manner. In a parallel experiment, we also tested the effect of both unconjugated and conjugated nanoparticles on virus neutralization. In this assay, the nanoparticles were pre-mixed with the virus in a dosage-dependent manner for an hour before challenging the target HeLa cells. The mock-treated virus and the mock-infected HeLa cells were kept as positive and negative controls, respectively. As shown in Figure 6B, quercetin-conjugated pAuNPsQ and cAuNPsQ retained the anti-HSV-1 activity. Interestingly, both quercetin-conjugated pAuNPsQ and cAuNPsQ showed more significant blockage than in the cell treatment assay (Figure 6A). Since quercetin-conjugated pAuNPsQ and cAuNPsQ displayed higher antiviral potency between 10–20 µg/mL, we decided to generate Ic50 value for both the pAuNPsQ and cAuNPsQ using reporter beta-galactosidase based HSV-1 entry assay. In this assay, HSV-1 (gL86) was exposed to the conjugated nanoparticles independently using a serial dilation (10–0.01 µg/mL) method before challenging to target HeLa cells. The sigmoidal curve (Figure 6C) generated from this experiment showed that pAuNPsQ and cAuNPsQ had IC50 of 4.33 µg/mL and 2.88 µg/mL, respectively.


**Preincubation of target cells with quercetin-conjugated gold nanoparticles (pAuNPsQ and cAuNPsQ) inhibits HSV-1 replication in a dosage-dependent manner.**


After cell entry, HSV-1 spreads to the neighboring host cells by cell-to-cell fusion [6]. During this process, HSV-infected cells express the viral envelope glycoproteins required for fusion on their surface, allowing the infected cell to bind and fuse with neighboring uninfected cells expressing target cell receptors. This results in the formation of large multinucleated giant cells called syncytia. A plaque reduction assay was performed to determine if the quercetin-conjugated pAuNPsQ and cAuNPsQ could also impact post-viral entry events such as productive virus replication. Productive replication of HSV-1 was determined by counting the number of plaques in the Vero cells in the presence and absence of quercetin-conjugated or unconjugated nanoparticles. In this experiment, a standard Vero cell plaque assay was performed. The cells were pre-treated with unconjugated or quercetin-conjugated nanoparticles for 60 min before challenging the cells with the replication-competent HSV-1 syncytia forming strain, KOS-804, at a multiplicity of infection (MOI) of 0.1. Untreated Vero cells served as a positive control, while untreated, uninfected Vero cells were used as a negative control in the experiment. Plaques were counted 48 h post-infection after fixing and staining cells with Giemsa. As indicated in Figure 7A, the quercetin-conjugated pAuNPsQ and cAuNPsQ severely impaired the plaque size and numbers in a dosage-dependent manner. Visual imaging of the plaques in the pre-treated cells with the quercetin-conjugated nanoparticles shown in Figure 7B demonstrates the dosage-dependent inhibitory effect of quercetin analogs on virus replication. At the same time, a similar concentration of unconjugated analogs of nanoparticles did not affect virus replication, suggesting the quercetin analogs were effective not only in blocking viral entry but also had a profound effect in affecting virus replication.


**The dosage-dependent effect of quercetin-conjugated pAuNPsQ and cAuNPsQ in a post-infection treatment model on HSV-1 replication.**


A post-infection treatment setting was investigated to determine if pAuNPsQ and cAuNPsQ had any effect on an established infection. In this experiment, target Vero cells were first infected with the virus for two hours, followed by exposure to the quercetin-conjugated nanoparticles in overlay media in a dosage-dependent manner. As indicated in Figure 8A, the post-treatment also significantly decreased the number of plaques in a dosage-dependent manner compared to the mock-treated positive control. Upon visualization in Figure 8B, plaque numbers and sizes were reduced, even in a post-infection treatment model. These results were impressive as they support the multifunctional properties of quercetin-conjugated nanoparticles affecting multiple stages of virus infection and under varying conditions of infection. The observed results demonstrate that blocking HSV-1 entry and replication by plant-derived nanoparticles can have both prophylactic and therapeutic applications.

## 3. Discussion

We used gold nanoparticles generated through intracellular fabrication in plant cell culture (Figure 1) and purified by the Sephadex G100 column chromatography in this viral inhibition study. The eluted fraction from the column with an absorption peak around 550 nm contained the desired mass of AuNPs for this study. The appearance of a single 550 nm peak in our gold solution points to the purity of pAuNPs, as also reported earlier in biogenic synthesis of gold nanoparticles [19,33]. The peak around 565 nm for the quercetin-coated particles (pAuNPQ) correlates with the increased size of the coated nanomaterial. Our previous study [19] also guided us that the size and shape of AuNPs depend on the concentration of KAuCl_4_ fed to the suspension medium used in alfalfa cell culture. We, thus, used the previously standardized condition (KAuCl_4_: 50 ppm; pH 5.7) of the plant cell culture [19] to obtain a mass of spherical nanoparticles in the size range of 20–22 nm. Using the DLS method of measurement in the present inhibition study, we again obtained the hydrodynamic diameter of pAuNPs over 21 nm. The significant increase in the hydrodynamic diameter of pAuNPsQ (43.82 nm) can be seen as a result of quercetin binding with the pAuNPs surface chemical groups. As knowledge about the nanoparticle surface electrical properties is critical in probing the interaction of particles with cells or organisms, we determined Zeta potentials where ζ value for pAuNPs was recorded as −23.5 mV (Figure 3A), which further dropped to −28.6 mV in quercetin coated gold nanoparticles (pAuNPsQ) (Figure 3B). It is important to note that the synthesis of pAuNPs occurred within the plant cell (cytoplasm)––without adding any external capping agent––where particles bonded with several functional groups, some carrying negative charges (e.g., hydroxyl, alkyl groups), present in the cytoplasmic environment. Furthermore, we coated those pAuNPs with quercetin that itself carries as many as five –OH groups and some carboxylic group per molecule. Thus, the acidity increased in these anionic nanoparticles, further reducing the ζ potential value to −28.6 mV, as also supported by other studies [34,35]. Evidence of the functional groups attached to the surface of pAuNPs can be seen in the FTIR analysis (Figure 3C,D). We also recorded that a shift in the hydroxyl peak of pAuNPs occurred from 2977 cm^−1^ to 2972 cm^−1^ in quercetin-coated gold (pAuNPsQ), indicating the conjugation of quercetin by the hydroxyl-amino linkage (Figure 3C,D). Furthermore, the conjugation of quercetin with particle surface functional groups also ensures good dispersion stability of these nanocomposites, as supported by ζ potential values [35].

Viral infections start with the attachment to host cells, usually by binding to the target receptor. If nanoparticles can effectively inhibit the attachment, then host cells will be free from infection. The current understanding of nanoparticle interactions with biological targets has transformed medical research, including the development of antiviral drugs [36,37,38]. Functional nanoparticles are being extensively investigated for their viricidal activity due to their large surface-to-volume ratio, high surface reaction activity, and size-dependent optical and electronic properties. Toward this goal, Gold (Au) and Silver (Ag) nanoparticles have been shown to exert inhibitory effects on viral infectivity and spread of multiple medically relevant viruses [39,40,41]. Specifically, silver nanoparticles capped with mercaptoethane (Ag-MES) and gold nanoparticles capped with mercaptoethane sulfonate (Au-MES NP) were shown as effective inhibitors of HSV-1 infectivity [42,43]. The high specificity of the Au and Ag nanoparticles to bind with HSV-1 paves the way for developing a new and highly effective microbicidal agent against chronic HSV infections. However, given the high costs of Au and Ag, new and cheaper alternatives are highly desirable. At the same time, it is also reported that various nanomaterials interact with human cells and macromolecules, producing inflammatory and other harmful effects [44]. Nanomaterials have also been shown to cross the blood-brain barrier (BBB), which might not always be intended and can cause inflammation at the BBB [45]. A strategy to counteract the undesirable effects may be to facilitate binding of nanostructures to larger units (macromolecules) without affecting the specific characteristics that originate at the nanoscale. Toward this approach, our earlier work with the chemically synthesized ZnO micro-nano particles had shown high promise as prophylactic agents against both HSV-1 and HSV-2 viruses [3].

In this study, we took an entirely different approach by synthesizing gold nanoparticles using plant cell suspension culture (Figure 1) followed by the conjugation of gold nanoparticles to a phenolic flavonoid, quercetin. The quercetin-conjugated pAuNPsQ were utilized in the present investigation to study their effects on SARS-CoV-2 and HSV-1 entry into host cells in respect of the unconjugated pAuNPs control. In parallel, we also tested quercetin-conjugated and unconjugated, chemically synthesized nanoparticles (cAuNPs) to compare their roles in preventing viral entry. A dramatic decrease in infectivity was seen when host cells were pre-treated with pAuNPs and cAuNPs prior to transduction with SARS-CoV-2. A previous report using molecular docking analysis and the surface plasma resonance demonstrated that quercetin exhibits a strong binding affinity to the ACE2 receptor, which impaired the binding of the virus-spike glycoprotein to ACE2 [46]. In the same study, the computational modeling also suggested that quercetin could bind to the spike glycoprotein’s RBD domain, implying that quercetin could also neutralize the virus [46]. Since the effect of pAuNPsQ and cAuNPsQ on virus neutralization was not observed in the case of SARS-CoV-2 (Figure 3B), while the neutralization effect was significantly higher in the case of HSV-1 (Figure 4B), this suggests nanoparticles may have displayed specificity for virus-specific envelope proteins. In contrast, our previous work with ZnO nanoparticles demonstrated a virostatic activity with neutralization of multiple viruses [47]. Future studies will be needed to discern if pAuNPsQ and cAuNPsQ- mediated inhibition of SARS-CoV-2 cell entry affected cell surface expression of two potentially key receptors—ACE-2 and heparan sulfate [7,8,9].

This study demonstrates that quercetin-conjugated pAuNPsQ and cAuNPsQ had a more robust response against HSV-1 compared to SARS-CoV-2 virus. The effectiveness of their treatments was not only confined to preventing HSV-1 entry, rather it also provided significant therapeutic efficacy by reducing the number of HSV-1-induced plaques formed both during pre- and post-infection treatment models (Figure 7 and Figure 8). However, the most direct way to suppress virus infection is to inactivate viruses either by binding molecules/agents to viral envelope glycoproteins or by changing their capsid protein structure to dramatically reduce the virulence. The application of nontoxic concentrations of pAuNPsQ and cAuNPsQ allowed us to conclude that the results were largely due to the inhibitory properties of the nanoparticle conjugate in a cell infection system. On the other end, both pAuNPsQ and cAuNPsQ agents inhibited entry of SARS-CoV-2 as well as HSV-1 entry during the cell pre-treatment experiment (Figure 7). Our specifically designed nanoparticles are speculated to possess a pleiotropic effect since the cell entry receptor requirements for SARS-CoV-2 and HSV-1 are significantly different except for the shared cell surface heparan sulfate receptors. It will be interesting to examine further if the above nanoparticle agent blocked viral entry because of their interactions with charge-based heparan sulfate—a unique receptor shared between both viruses [8,14]. Furthermore, the greater efficacy of quercetin-conjugated pAuNPs in the case of both viruses indicates that our uniquely designed nanomaterial can be also used as a drug delivery vector to counteract a viral infection.

One limitation in our study was the use of pseudovirus SARS-CoV-2, which is replication incompetent and only expresses spike glycoprotein. Since quercetin has been shown to interact also with the N protein of SARS-CoV-2 [48], future studies will be needed to verify the antiviral effect of quercetin-conjugated pAuNPsQ and cAuNPsQ nanoparticles using authentic SARS-CoV-2 virus under BSL-3 conditions. Our study used the uniquely prepared nanomaterials (generated intracellularly) from the cell suspension culture of *Medicago sativa*, which allows novel macromolecular coatings over the nanoparticles depending on the internal cellular environment of the culture. While this was useful in laying the groundwork, exploring plant cell culture of notable medicinal plants would be advantageous to develop novel nanomaterials or nanocomposites that may carry secondary metabolites of interest in the generation of valuable antiviral agents [49,50,51,52]. The other advantage of standardizing such a system would be the mass production of novel antiviral particles with reduced biohazard risk that could serve as an eco-friendly intervention to treat viral infections. Finally, our work over the years has generated panels of anti-HSV-1 envelope glycoprotein D (gD) and anti-heparan sulfate peptides [53,54,55], which also offers equally an exciting possibility to conjugate the antiviral peptides to the plant cell derived nanoparticles to target specific viruses (Figure 9). Similarly, the opportunity to test anti-adhesive and anti-bacterial biomaterials [56], especially against oral herpesvirus remains a valid option.

## 4. Material and Methods

**Establishment of Alfalfa (*Medicago sativa*) Cell Culture.** *M. sativa* (alfalfa) seeds were purchased from Athens seed company (Watkinsville, GA, USA). *M. sativa* hypocotyl regions were dissected out from one-week-old, aseptically grown seedlings and inoculated on Murashige and Skoog (MS) medium, supplemented with 30% sucrose, 0.8% agar, 0.2 mg/L indole–3–acetic acid (IAA), and 0.2 mg/L 6-Benzylaminopurine (BA) following our own protocol [19]. After two weeks, the proliferated callus mass was used to establish a shaking cell suspension culture. Following an incubation of 5–7 days, the cell suspension was collected, discarding the clumps of callus remaining in the bottom. The single-cell nature of the culture was confirmed by microscopic examination. Cultures were maintained by sub-culturing every two weeks by adding fresh medium following the removal of the old one.

**Synthesis & Purification of Plant-derived Gold Nanoparticles (pAuNPs).** The above *M. sativa* suspension cells were used to facilitate the fabrication of gold nanoparticles following the methods of Bhaskaran et al. [19]. Potassium gold (III) chloride (KAuCl_4_; ThermoFisher, Waltham, MA, USA) was supplemented to the cell suspension and incubated on a shaker for 24 h in darkness. Following the incubation, the original yellow color of the gold chloride solution turned to a ruby red color as a result of KAuCl_4_ reduction, pointing to the formation of gold nanoparticles. Aliquots of the culture were then collected and centrifuged for 2 min (14,000 rpm), and the medium was removed. After repeated washing, cells were further sonicated and centrifuged to separate gold nanoparticles from the cell suspension. The extracted gold nanoparticles were further purified by the Sephadex G100 column following the methods described by Bhaskaran et al. 2019 [19].

**Coating of Plant-derived and Chemically Synthesized Gold Nanoparticles with Quercetin.** Various concentrations of quercetin (ThermoFisher, Waltham, MA, USA) were added to a fixed concentration (20 µg/mL) of gold nanoparticles aqua solution and agitated for 2 h at room temperature to facilitate the coating of gold nanoparticles with quercetin. After a lapse of 2 h, the reaction mixture was incubated at 4 °C for 12 h. The quercetin-coated plant cell-derived gold nanoparticles (pAuNPsQ) were then washed with sterile water multiple times to elute the unreacted quercetin. After the final rinsing, sterile water (pH 6.5) was added to the nanomaterial–quercetin complex. Specific concentrations (as shown in Results and figures) of quercetin-coated gold nanoparticles were used in viral inhibition assays. A chemically synthesized amine functional group capped with gold spherical nanoparticles (cAuNPs; 20–22 nm size) was purchased from Nanopartz. Inc (Loveland, CO, USA). The coating process with the quercetin to the cAuNPsQ was identical to the pAuNPsQ as described above.

### 4.1. Characterization of Gold Nanoparticles

**UV Measurement Spectroscopy.** UV-visible spectrophotometer (Jason V-730, Jasco Inc., Easton, MD, USA) was used to determine the absorbance spectrum of prepared gold nanomaterials. For UV measurement, 1 mL of diluted nanoparticles solution was collected in the cuvette for optical density measurement under the UV-visible optical range.

**Particle Size and Zeta (ζ) Potential Measurement of Gold Nanoparticles.** Particle size and zeta potential of prepared nanoparticles were determined by Dynamic Light Scattering (DLS) instrument (Zetasizer Nano-ZS, model: ZEN3600, Malvern Panalytical Ltd., Malvern, UK) with He-Ne laser beam (633 nm, fixed scattering angle 90°). Nanomaterials were analyzed at 25 °C with 1 mL aliquot of the nanoparticle’s solution collected in a cuvette. Size and zeta potential values were measured and displayed in the results as the average value of three runs.

**Fourier-transform infrared spectroscopy.** Fourier-transform infrared spectroscopy (FTIR, Nicolet iS5, Thermo Scientific, Waltham, MA, USA) measurement was performed to determine the functional group present in the prepared nanoparticles. In this measurement, a drop of sample was placed in the instrument for scanning under attenuated total reflection (ATR) mode.

### 4.2. Cells and Viruses

Human cervical epithelial cells (HeLa, ATCC, Manassas, VA, USA), African green monkey kidney cells (Vero, ATCC, Manassas, VA, USA), and Human kidney epithelial cells stably expressing ACE2 receptor (HEK293T-ACE2, Genecopoeia, Rockville, MD, USA) were used for in vitro studies. Cells were grown at 37 °C with 5% CO_2_ and were passaged according to the manufacturer’s recommendations using 0.5% trypsin EDTA after reaching near-confluence. Cells were grown in Dulbecco’s modified Eagle medium (DMEM) supplemented with 100 U/mg/mL P/S, and 10% FBS. HEK293T-ACE2 cells were grown in DMEM supplemented with 100 U/mg/mL P/S, 10% FBS, and 100 µg/mL hygromycin B. Spike-pseudotyped lentivirus particle having D614G mutation (Cat # SP003-100) was purchased from Genecopoeia (Rockville, MD, USA) and was used for luciferase-based studies. This study also used wild-type HSV-1 KOS strain (ATCC, Manassas, VA, USA, VR-1493) and the recombinant β-galactosidase-expressing reporter HSV-1 KOS (gL86) strain. Using complementing Vero cell lines, viral stocks were propagated, while the viral titers were determined by standard plaque assay, and the resulting stocks were stored at −80 °C.

### 4.3. LDH Cytotoxicity Assay

Cellular toxicity of the gold nanoparticles used in this study was assessed using a lactate dehydrogenase (LDH) cytotoxicity assay kit (Pierce Biotechnology, Rockford, IL, USA) per the manufacturer’s recommendations. HeLa and 293T-ACE2 cells were plated (15,000 cells/well) in triplicate and grown overnight in complete media, followed by dosage-dependent treatment with pAuNPsQ and cAuNPsQ. Then, 10× lysis buffer was used for maximum LDH activity controls, while 10 µL sterile water was used as a mock treatment for the spontaneous LDH release controls. Following incubation at different time points, 50 µL supernatant was combined with 50 µL reaction mixture, and the plate was incubated at room temperature for 30 min. The LDH activity was determined by measuring the absorbance at 492 nm and 620 nm on a microplate photometer and subtracting the 620 nm values from the 492 nm values to remove background.

### 4.4. SARS-CoV-2 Spike Protein-Pseudotyped Lentivirus Entry Assay

HEK293T-ACE2 cells were grown to 70–80% confluence in a 96-well plate in Dulbecco’s Modified Eagle Medium (DMEM) supplemented with 10% heat-inactivated fetal bovine serum (FBS). The following day, growth media was removed and replaced with serum-free Opti-MEM™ media containing pAuNPsQ and or cAuNPsQ (10 µg/mL or 20 µg/mL) for 1 h at 37 °C for cell-treated experiments. Treated cells were then transduced or infected with 2 × 10^8^ RLU/mL SARS-CoV-2 pseudovirus per well and moved to 4 °C for 2 h to enhance virus binding. For virus-treated experiments, the volume equivalent of 2 × 10^8^ RLU/mL pseudovirus per well was treated for 1 h with dilutions of each pAuNPsQ and cAuNPsQ (10 µg/mL or 20 µg/mL) or mock-treated with PBS. After treatment, the virus and pAuNPsQ and or cAuNPsQ preparations were diluted with complete media, and the plate was incubated for 2 h at 4 °C. Following cold incubation, the infected plates were moved to 37 °C for 48 h. After 48 h, the media was removed, and 30 µL/well of reporter lysis buffer (Promega, Madison, WI, USA) was added. The cells were scraped and freeze-thawed at −80 °C to complete cell lysis. 20 µL of lysate was transferred to a white 96-well plate, 50 µL of luciferase assay reagent (Promega, Madison, WI, USA) was added, and luciferase activity was recorded using the EnSpire Multimode Plate Reader (PerkinElmer, Waltham, MA, USA) at a speed of 0.5 s per well. 

### 4.5. Herpes Simplex Virus Typ-1 (HSV-1 gL86) Viral Entry Assay

HeLa cells were grown to 80–90% confluence in 96-well plates at 37 °C with 5% CO_2_ overnight. The following day, cells were washed with PBS and pre-treated with Opti-MEM containing pAuNPsQ and or cAuNPsQ dilutions (10 µg/mL or 20 µg/mL) for 1 h at 37 °C with 5% CO_2_ for cell-treated experiments. Treated cells were then infected with HSV-1 KOS (gL86) in Opti-MEM at a multiplicity of infection (MOI) of 5 at 37 °C with 5% O_2_. Cells were incubated with the inoculum for 6 h at 37 °C with 5% CO_2_. For virus-treated experiments, 5 MOI HSV-1 KOS (gL86) was incubated with 10 µg/mL or 20 µg/mL pAuNPsQ and or cAuNPsQ dilution in serum-free media for 1 h at room temperature with shaking before adding to cells. Infected, untreated cells served as a positive control, while untreated, uninfected cells were used as a negative control in both experiments. After 6 h, the viral inoculum was removed, cells were washed 3× in PBS, and 50 µL substrate solution (3.0 mg/mL ONPG, 0.6% NP40 in PBS) was added. The plate was incubated at 37 °C with 5% CO_2_ for 1 h and was read at 405 nm on the Multiskan FC Microplate Photometer (ThermoFisher, Waltham, MA, USA). GraphPad Prism 9 was used to generate graphs, analyze results, and determine IC_50_ values.

### 4.6. Plaque Reduction Assay

Vero cells were grown to confluence overnight in a 12-well plate and pre-treated with pAuNPsQ and or cAuNPsQ in a dosage-dependent manner (10, 7.5, 5, 2.5 µg/mL) for 1 h at 37 °C in 5% CO_2_ for pre-treatment experiments. Following treatment, cells were incubated with a 0.1 MOI HSV-1 (KOS-804) syncytial mutant [57] for 2 h at low speed on a rocker. The virus inoculum was removed, and cell monolayers were washed 3× with PBS. Following infection, cells were incubated for 48–72 h in 1.5 mL overlay media containing DMEM supplemented with 1% heat inactivated FBS and 0.5% methyl cellulose. For post-infection treatment, cells were infected for 2 h with 0.1 MOI HSV-1, washed 3× in PBS, and overlaid with 1% heat-inactivated FBS and 0.5% methyl cellulose containing dilutions of nanoparticles (10, 7.5, 5, 2.5 µg/mL). Infected, untreated cells served as a positive control, while untreated, uninfected cells were used as a negative control in both experiments. After 48 h, when the cytopathic effect was observed, cells were washed 3× in PBS and fixed for 30 min in 4% paraformaldehyde. Cell monolayers were then stained with Giemsa (Sigma, St. Louis, MO, USA), and plaques were counted under 10× magnification. Images of plaques were captured using the EVOS FL auto imaging system (ThermoFisher, Waltham, MA, USA), under 4× magnification.

### 4.7. Statistical Analysis

All experiments were performed in triplicates unless otherwise stated to confirm the consistency of results. GraphPad Prism 9 was used to analyze data for statistical significance (*p* < 0.05). One-way analysis of variance (ANOVA) was used to determine significance between control and experimental groups followed by Bonferroni’s multiple comparisons test to determine significance compared to the positive control. In all figures, columns represent the mean of the data collected and error bars represent SD.

## Figures and Tables

**Figure 1 ijms-24-14792-f001:**
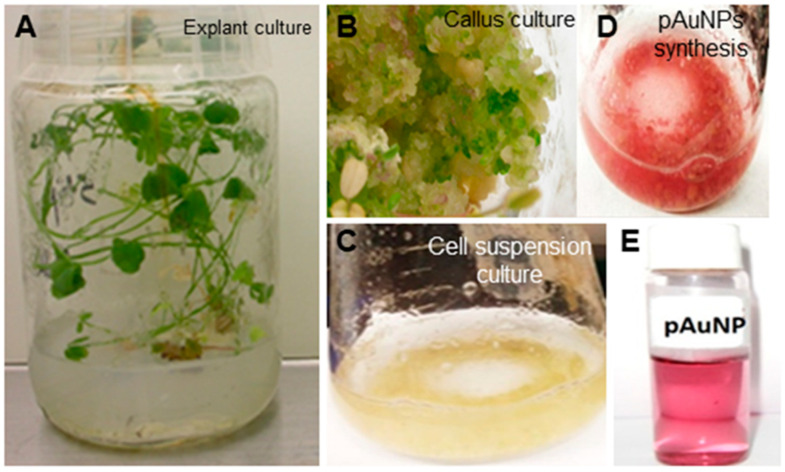
Synthesis of plant cell derived gold nanoparticles (pAuNPs). *Medicago sativa* (Alfalfa) in vitro plantlets were established (panel (**A**)) followed by the callus mass (panel (**B**)) to generate cell suspension culture (panel (**C**)). Upon feeding the cell suspension culture with the salts of KAuCl4 resulted in the formation of gold nanoparticles (panel (**D**)). The gold nanoparticles were separated from the cell suspension culture using sonication and centrifugation. The extracted gold nanoparticles were further purified by the sephadex G100 column (panel (**E**)). The coating of plant cell-derived gold nanoparticles with quercetin is described in Section 4.

**Figure 2 ijms-24-14792-f002:**
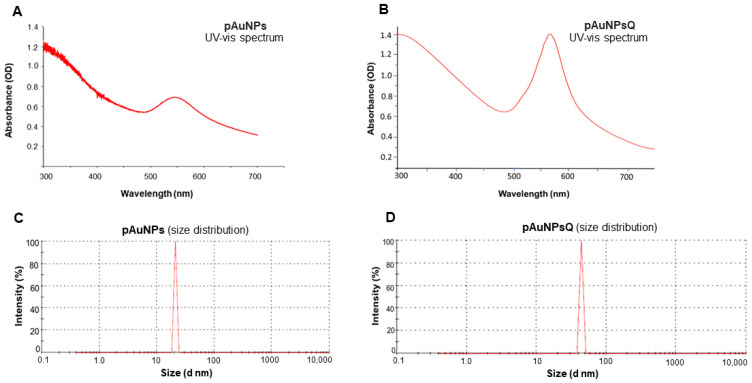
Characterization of plant cell derived gold nanoparticles (pAuNPs) with and without quercetin. The images showing the characterization of nanoparticles were carried out by measuring optical density under UV-visible spectrophotometer in presence and absence of quercetin (panels (**A**,**B**)). The characterization of nanoparticles for the particle size and distribution in the presence and absence of quercetin (panels (**C**,**D**)) is shown.

**Figure 3 ijms-24-14792-f003:**
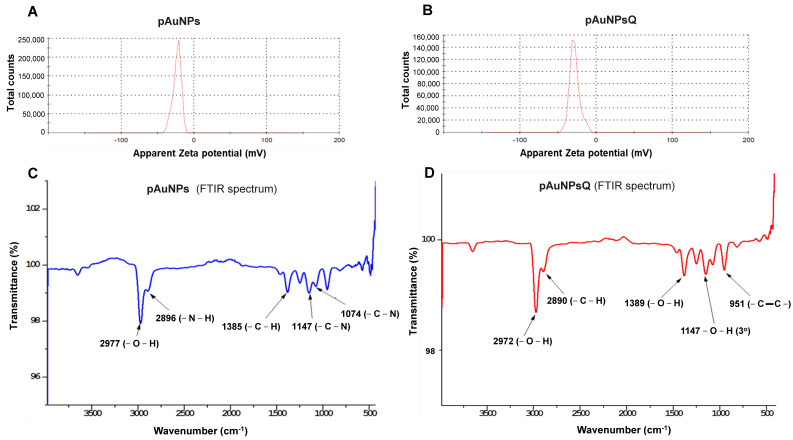
Characterization of plant cell derived gold nanoparticles (pAuNPs) with and without quercetin. The images show the characterization of nanoparticles for the zeta potential by Dynamic Light Scattering (DLS) in presence and absence of quercetin (panels (**A**,**B**)) instrument in a triplicate experiment. Further, the analysis of functional groups present in the plant cell-engineered gold nanoparticles with and without quercetin was carried out by Fourier-transform infrared spectroscopy (FITR; panels (**C**,**D**)).

**Figure 4 ijms-24-14792-f004:**
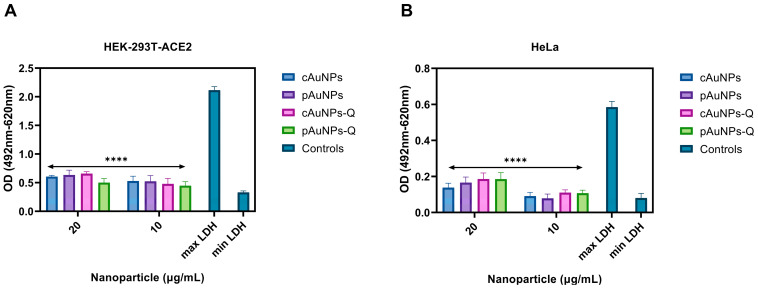
The effect of plant cell derived gold nanoparticle (pAuNPs) and chemically synthesized gold nanoparticles (cAuNPs) conjugated with and or without quercetin on the cellular toxicity using LDH assay. (**A**). HEK-293T cells that stably express ACE-2 receptors were incubated in the presence or absence of the nanoparticles (NPs) at 20 µg/mL and 10 µg/mL for 48 h to mimic the conditions used during SARS-CoV-2 cell entry. (**B**). HeLa cells were incubated in the presence and absence of NPs at 10 µg/mL and 20 µg/mL for 6 h to mimic the conditions used during the HSV-1 cell entry. The cells treated with media alone represent the spontaneous LDH release (min LDH), while cells lysed with detergent represent maximum LDH release. Asterisks (****) indicate a significant difference between the maximum LDH release and the nanoparticle-treated cells (*p* < 0.0001).

**Figure 5 ijms-24-14792-f005:**
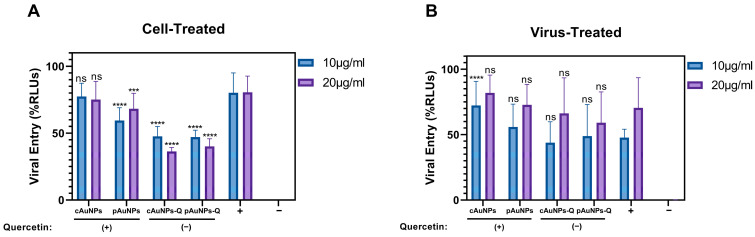
Pretreatment of target cells with pAuNPsQ and cAuNPsQ affects SARS-CoV-2 pseudovirus cell entry, but pretreatment of pseudovirus does not affect cell entry. (**A**). Monolayers of HEK-293T cells were preincubated with pAuNPs or cAuNPs with or without quercetin at 10 µg/mL and 20 µg/mL before transducing the target cells with SARS-CoV-2 (D614G) pseudovirus. (**B**). SARS-CoV-2 pseudovirus was pre-treated with 10 µg/mL or 20 µg/mL of gold nanoparticles followed by transduction of the target cell with the nanoparticle-treated pseudovirus. The untreated HEK293T-ACE2 cells transduced or infected with the SARS-CoV-2 pseudovirus were considered as a positive control (+), while the un-transduced HEK293T-ACE2 cells were considered as a negative control (−). Asterisks indicate a significant difference between the positive control and the cells treated with AuNPs (**** *p* < 0.0001, *** *p* < 0.0003, “ns” indicates not significant).

**Figure 6 ijms-24-14792-f006:**
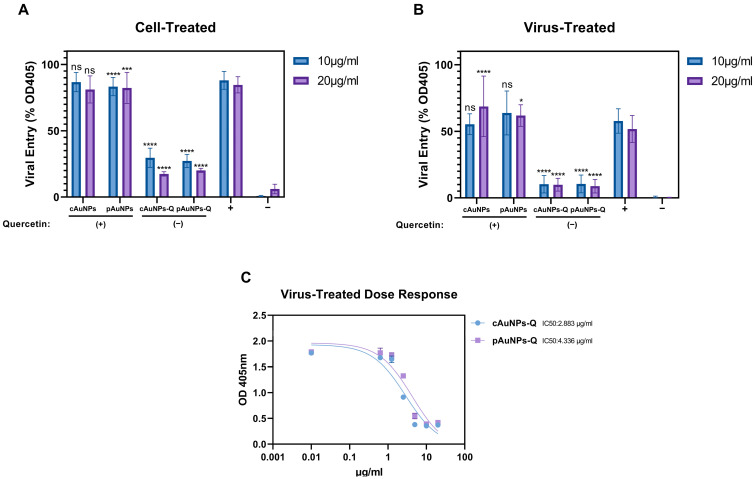
Pretreatment of target cells and HSV-1 with pAuNPsQ and cAuNPsQ significantly impairs herpes simplex virus type-1 (HSV-1) cell entry. (**A**). HeLa cells were preincubated with pAuNPs and cAuNPs with or without quercetin at 10 µg/mL and 20 µg/mL followed by infection with HSV-1 (gL86). (**B**). HSV-1 was pre-incubated with 10 µg/mL and or 20 µg/mL of gold nanoparticles followed by infection of the target HeLa cell with the nanoparticle-treated virus. (**C**). The dose-response curve generated for the determination of IC_50_ values of pAuNPsQ and cAuNPsQ against herpes simplex virus type-1 entry. In all viral entry experiments, the untreated HeLa cells infected with the HSV-1 were considered as a positive control (+), while the uninfected HeLa cells were considered as a negative control (−). Asterisks indicate a significant difference between the positive control and the cells treated with AuNPs (* *p* ≤ 0.05, *** *p* ≤ 0.0001, **** *p* ≤ 0.0001, ns not significant).

**Figure 7 ijms-24-14792-f007:**
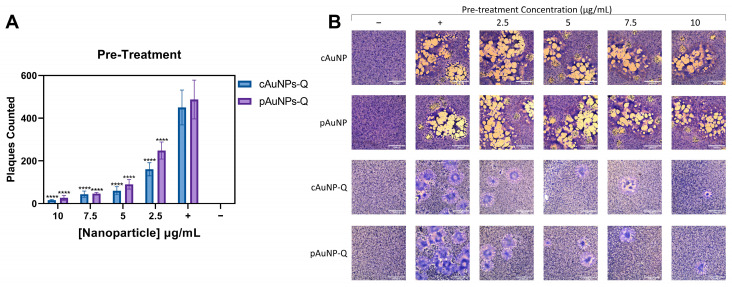
Prophylactic effects of pAuNPsQ and cAuNPsQ against herpes simplex virus type-1 replication and syncytial formation. The effect of pre-treatment with quercetin-conjugated pAuNPsQ and cAuNPsQ on viral spread and replication was determined by syncytial plaque assay in Vero cells. (**A**) Dosage-dependent effect of pAuNPsQ and cAuNPsQ on plaque formation in HSV-1 KOS (804) infected cells under various treatment conditions as indicated. (**B**) Visual reduction in plaque number and sizes are shown in the panels. Uninfected, untreated cells used as negative control (−), infected, untreated cells used as positive control (+). The data represent observations from three independent experiments. Scale bar: 1000 µm. Asterisks indicate a significant difference between the positive control and the cells treated with AuNPs (**** *p* ≤ 0.0001).

**Figure 8 ijms-24-14792-f008:**
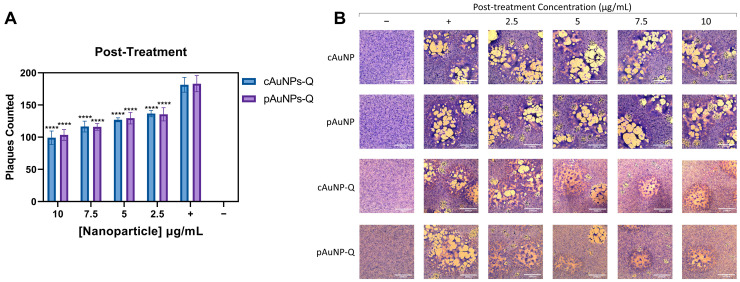
Therapeutic effects of pAuNPsQ and cAuNPsQ in a post-HSV-1 infected cell model. The post-entry effect of pAuNPsQ and cAuNPsQ added to the target Vero cells post-infection with HSV-1 KOS (804) strain on viral replication was determined by plaque quantification. (**A**) Dosage-dependent effect of pAuNPsQ and cAuNPsQ on plaque formation in HSV-1 KOS (804) infected cells under various treatment conditions as indicated. (**B**) Visual reduction in plaque number and sizes are shown in the panels. Uninfected untreated cells used as negative control (−), infected untreated cells used as positive control (+). The data represent observations from three independent experiments. Scale bar: 1000 µm. Asterisks indicate a significant difference between the positive control and the cells treated with AuNPs (**** *p* ≤ 0.0001).

**Figure 9 ijms-24-14792-f009:**
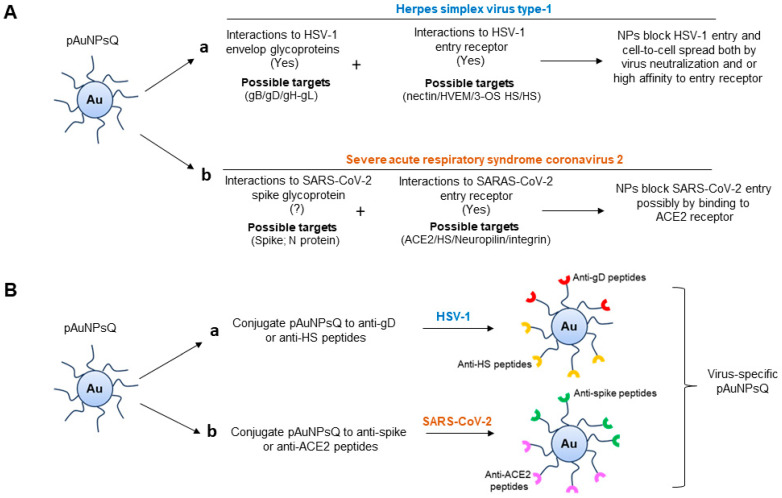
Possible mechanism by which plant cell derived gold nanoparticles conjugated to quercetin (pAuNPsQ) inhibits virus–host cell interactions. (**A**) Quercetin-conjugated gold nanoparticles block HSV-1 entry by interacting with the HSV-1 entry glycoprotein and/or targeting host cell receptors, as evident from our virus neutralization and cell pre-treatment assays (sub panel (**a**)). Since multiple viral glycoproteins and host cell receptors are involved in facilitating HSV-1 entry, the actions by pAuNPsQ could be predicted to be pleiotropic in nature, having an affinity for one or more HSV-1 envelope virus glycoproteins (gB, gD, gH-gL) as well as for one or more host cell receptors. In contrast, quercetin-conjugated gold nanoparticles did not affect SARAS-CoV-2 neutralization, while pre-treatment with quercetin conjugated gold nanoparticles inhibited SARS-CoV-2 entry suggesting the possible interactions between pAuNPsQ with the host cell surface ACE2 receptor (sub panel (**b**)). (**B**) The proposed development of virus-specific gold nanoparticles can be generated by conjugating pAuNPsQ to the virus-specific anti-envelope glycoprotein D (gD) peptides in case for HSV-1 (sub panel (**a**)). Similarly conjugating pAuNPs to anti-spike envelope proteins could be used for the development of robust anti-SARS-CoV-2 agent (sub panel (**b**)). In addition, tagging quercetin-based gold nanoparticles with anti-host cell receptor peptides (anti-heparan sulfate or anti-ACE2) can equally be tailored for their antiviral activity against specific viruses.

## Data Availability

The research data will be made available upon request to the corresponding author.

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
