# Peer review of "Plant Cell-Engineered Gold Nanoparticles Conjugated to Quercetin Inhibit SARS-CoV-2 and HSV-1 Entry"

_ijms, 2023, doi:10.3390/ijms241914792_

Round 1

Reviewer 1 Report

The work by J. Elste et al. describes the engineering and the antiviral activity of gold nanoparticles and their conjugates with a flavonoid compound (quercetin). The manuscript presents a comparative study of plant cell-engineered and chemically synthesized gold nanoparticles inhibiting the entry of 2 model viruses into the cells. The manuscript includes information on gold nanoparticle preparation and estimation of its antiviral activity. However, the authors should provide additional data on the characterization of the nanomaterials designed. Below are some points that could help improve this work's impact.

  1. What advantages suggest the gold nanoparticles engineered in this study compared to the gold or silver nanoparticles exhibiting antiviral activity already reported, e.g., https://doi.org/10.1016/j.bbrc.2020.09.018https://doi.org/10.3390/molecules26195960https://doi.org/10.3390/microorganisms10010110
  2. After being synthesized, nanoparticles were purified using column chromatography (p. 12 lines 429 - 431). How the purity of the resultant particles was controlled in this study?
  3. The particle size was estimated using dynamic light scattering analysis (p. 2-3 lines 95-99, p.13 lines 449-454). This technique provides only the information on nanoparticle hydrodynamic diameter. However, the physical size of the nanoparticle (e.g., measured by TEM) may significantly vary from the hydrodynamic diameter values. The authors also claim nanoparticle shape as one of the characteristics to be controlled in the study (p. 1 lines 25-26 «…we also tested the chemically synthesized gold nanoparticles 24 (cAuNPs) of identical size and shape…» ). What were the physical size and shape of the gold nanoparticles analyzed in the study? Are these characteristics also likely to influence the antiviral activity of these particles?  Please provide some results on the characterization of nanoparticles, including particle shape and physical size, both for plant cell-engineered and chemically synthesized samples. 
  4. In this study, the gold nanoparticles were modified using quercetin through the electrostatic interactions. How was the quantity of quercetin bound with the nanoparticle surface controlled? Please specify this issue.
  5. The nanoparticles were introduced into the cell culture medium supplemented with FBS to investigate antiviral activity and cytotoxicity. Was the nanoparticle colloidal stability verified in the complex cell culture media? 
  6. The figure captions (p.5-7) are overloaded with technical information; the figure captions should be updated and presented more explicitly. 
  7. What equipment was used to monitor the reduction in plaque number and sizes (the data presented in Fig. 5 and Fig. 6). Provide the details on the equipment within the «Materials and Methods» section. The images in panel B (Fig. 5 and 6, p.8, p. 9) should be provided with scale bars. 
  8. Figures 1-4 should be corrected:
    • Fig.1, panels B and C are low quality: numbers, letters in the names of the axes, and the distribution curves are too small and, thus, are difficult to read in the current version.
    • Present Fig. 2 in color and provide it with a color legend.
    • Increase the size of the symbols, indicating the statistical differences between the samples presented in Fig. 3 and 4.   

Author Response

September 19th 2023

Editor
International Journal of Molecular Sciences
Re: Manuscript ID: ijms-2588649

Dear Editor,

We would like to thank you and the reviewers for a prompt review of our manuscript. We are pleased to inform you that we have made the required revisions to the original manuscript by incorporating the required changes as suggested by the reviewers.  Below is the itemized description of changes made in the revised manuscript  following the reviewer’s suggestions.  In addition, we have further provided insight regarding the advantages of plant engineered nanoparticles in viral host adaptation and spread across species. All the changes made to the revised manuscript are highlighted in yellow.

Reviewer 1

  1. What advantages suggest the gold nanoparticles engineered in this study compared to the gold or silver nanoparticles exhibiting antiviral activity already reported, e.g., https://doi.org/10.1016/j.bbrc.2020.09.018, https://doi.org/10.3390/molecules26195960, https://doi.org/10.3390/microorganisms10010110? 

Response:  As mentioned by the reviewer 1, we have clearly stated the advantages of our plant cell engineered gold nanoparticles in the manuscript in the discussion section. In fact, our study is different from the referred - three previously published studies in respect of the nature and characteristics of gold nanoparticles, and the human cell lines that we used to test inhibitory potentials of AuNPs against both DNA (herpes simplex virus) and RNA (SARS-CoV-2) viruses. Here we show the effects of plant cell-generated novel gold nanoparticles, naturally carrying the functional groups such as alkyl, hydroxyl/amino groups present from the cytoplasmic environment of plant cell culture (intracellular fabrication details already published by our group: Bhaskaran et al., 2019 Scientific Reports). Our focus in this study is to allow these coated nanomaterials to conjugate with a food flavonoid (therapeutically important quercetin) as a possible delivery agent in the viral inhibition study. In parallel, we also used commercially available synthetic gold nanoparticles (AuNPs) only for comparison with our novel plant cell generated nanomaterial. Finally, the mode of action of plant cell-generated along with synthetic gold nanoparticles in presence and absence of quercetin was further examined on two medically important viruses: HSV-1 & SARS CoV-2 using receptor specific (HeLa, HEK293T-ACE2, and Vero) cell lines under pre- (for prophylaxis usage) and post-treatment (for therapeutic usage) model.

Categorical differences from the cited studies are mentioned below:

  • Jeremiah et al. 2020: Biophys Res Commu (https://doi.org/10.1016/j.bbrc.2020.09.018): This study tested the effect of silver (Ag) nanoparticles (1-10 ppm) on SARS-CoV-2 using VeroE6 & Calu3 cell lines. While in our study we have used biogenic 20-21 nm gold (AuNPs) nanoparticles (naturally carrying alkyl, hydroxyl/amino groups) to test their effects, as a drug (quercetin) delivery agent, on HSV-1 & SARS CoV-2 using receptor specific cell lines: HeLa, HEK293T-ACE2 and Vero. Our data highlights that antiviral mechanism of the plant engineered gold nanoparticles conjugated with flavonoids varies depending on the virus. The later results provide novel insight into mechanistic details of antiviral mechanism.
  • Paradowska et al. 2021: Molecules (https://doi.org/10.3390/molecules26195960): This study used non functionalized AuNPs (10, 16 nm) against HSV-1 in Vero cells and found that AuNPs were capable of reducing the cytopathic effect (CPE) of HSV-1 in Vero cells by interacting with viral envelop proteins; again our gold nanoparticles were plant cell-engineered conjugated with flavonoids which showed contrasting antiviral mechanism in HSV compared to SARS-CoV-2 virus – which has not been previously shown.
  • Krzyzowska et al. 2022: Microorganisms (https://doi.org/10.3390/microorganisms10010110): This study used antiviral and cytotoxic activities of lactoferrin-functionalized noble metal nanoparticles (LF-Ag/AuNPs) and tested in human skin HaCaT and vaginal VK-2-E6/E7 keratinocytes. As mentioned above, we have used biogenic 20-21 nm AuNPs (naturally carrying alkyl, hydroxyl/amino groups) to test their effects, as a drug (quercetin) delivery agent, on HSV-1 & SARS CoV-2 using receptor specific cell lines (HeLa, HEK293T-ACE2 and Vero) testing both the prophylaxis (pre- treatment) and therapeutic (post-treatment) antiviral potential of plant derived nanoparticles, which were previously not reported.

  1. After being synthesized, nanoparticles were purified using column chromatography (p. 12 lines 429 - 431). How was the purity of the resultant particles controlled in this study?

Response:  Based on our previous study (Bhaskaran et al. 2019; Scientific Reports), we had learned that the different fractions eluted from the gel column had different peaks under UV-Visible spectroscopy. We thus selected in the present experimental set up the eluted fraction from the gel permeation column showing a peak around 550 nm. The appearance of a single 550nm peak in our gold sample confirms the desired nature of AuNPs (20-22 nm); in case of impurity more than one peak might have emerged in the test solution.

  1. The particle size was estimated using dynamic light scattering analysis (p. 2-3 lines 95-99, p.13 lines 449-454). This technique provides only the information on nanoparticle hydrodynamic diameter. However, the physical size of the nanoparticle (e.g., measured by TEM) may significantly vary from the hydrodynamic diameter values. The authors also claim nanoparticle shape as one of the characteristics to be controlled in the study (p. 1 lines 25-26 «…we also tested the chemically synthesized gold nanoparticles 24 (cAuNPs) of identical size and shape…» ). What were the physical size and shape of the gold nanoparticles analyzed in the study? Are these characteristics also likely to influence the antiviral activity of these particles?  Please provide some results on the characterization of nanoparticles, including particle shape and physical size, both for plant cell-engineered and chemically synthesized samples. 

Response: We agree with the reviewer’s comment that Transmission Electron Microscopy (TEM) is the best method to determine the physical size of nanoparticles. However, our intention to use the Dynamic Light Scattering (DLS) analysis in the present experiment was to specifically determine the hydrodynamic diameter d(H), as the d(H) value gives the more realistic value of particles suspended in liquid while examining their interactions with cell receptors or viral spikes. It is not out of place to mention that our focus in this study is to test the antiviral activity by mixing the aqua solutions of biogenic AuNPs into the viral culture media.

Size & Shape: We also learned from our previous study, focused on the novel synthesis of AuNPs using the cell suspension culture of alfalfa (Bhaskaran et al. 2019), that the size and shape of AuNPs depend on the concentration of  KAuCl4 fed to alfalfa cell suspension medium. We thus used the previously standardized condition (KAuCl4: 50 ppm; pH 5.7) of plant cell culture to obtain a mass of spherical nanoparticles in the size range of 20-22 nm.  In the cited study, we had confirmed the size and shape of each column fraction using TEM-EDS (Bhaskaran et al. 2019). Using the DLS method in the present study, we again obtained the hydrodynamic diameter of nanoparticles  over 21nm (shown in the Results section under Size & Zeta Potential). Commercially available AuNPs of the known specifications (spherical 20 nm) were used for comparison in viral inhibition assay.

We  apologize that the figures showing size distribution and Zeta Potential values were not correctly numbered (the numbering was messed up because of combining all characterization parameters in one plate). We have now corrected the errors and separated them for clarity.

  1. In this study, the gold nanoparticles were modified using quercetin through electrostatic interactions. How was the quantity of quercetin bound with the nanoparticle surface controlled? Please specify this issue.

Response:  It is important to clarify that our cell suspension culture method of AuNPs fabrication was based on the reduction of fed KAuCl4  within the cellular environment of the culture. In other words, AuNPs were synthesized intracellularly in the Alfalfa suspension culture, as evidenced by TEM in our previous publication (Bhaskaran et al. 2019). Furthermore,  functional groups (such as alkyl, hydroxyl/amino groups) were present in the plant intracellular (cytoplasmic) environment bound with gold nanoparticles through various interactions. It is for this reason, we carried out the  Fourier-Transform Infrared Spectroscopy (FTIR) analysis to determine the chemical groups attached  to our nanomaterial. This way, our pAuNPs are unique and carry specific sites for quercetin binding. Our long-term goal is to apply these nanomaterials in drug delivery.

However, the query about the quantity of quercetin carried by pAuNPs is a pertinent question that requires the quantification of residues after quercetin treatment of nanoparticles, which we couldn't complete because of problems in the HPLC facility. However, we noticed significant fading in the color of quercetin-nanomaterial complex after incubation with AuNPs. In our viral inhibition study, we have used a control of pAuNPs without any quercetin, so the difference in inhibition caused by the quercetin containing AuNPs is significant.          

  1. The nanoparticles were introduced into the cell culture medium supplemented with FBS to investigate antiviral activity and cytotoxicity. Was the nanoparticle colloidal stability verified in the complex cell culture media? 

Response: The nanoparticles were introduced in these experiments in Opti-MEM, a serum free media to investigate antiviral and cytotoxicity activity. The nanoparticle colloidal stability was not performed.

  1. The figure captions (p.5-7) are overloaded with technical information; the figure captions should be updated and presented more explicitly. 

Response:  As suggested by the reviewer we have edited the figure legends by removing unnecessary technical information.

  1. What equipment was used to monitor the reduction in plaque number and sizes (the data presented in Fig. 5 and Fig. 6). Provide the details on the equipment within the «Materials and Methods» section. The images in panel B (Fig. 5 and 6, p.8, p. 9) should be provided with scale bars. 

Response:  Giemsa-stained cells showing plaques formation were monitored daily using a light microscope under 10x magnification until cytopathic effect was observed in the positive control. The materials and methods section has been updated with equipment details and 1000 µm scale bars have been added to the plaque images.

  1. Figures 1-4 should be corrected:
    • Fig.1, panels B and C are low quality: numbers, letters in the names of the axes, and the distribution curves are too small and, thus, are difficult to read in the current version.

Response:  As suggested by the reviewer we have provided the high-resolution panels for figure 1.

    • Present Fig. 2 in color and provide it with a color legend.

Response:  As suggested by the reviewer, all the figures were updated to include color coded columns along with the legend.

    • Increase the size of the symbols, indicating the statistical differences between the samples presented in Fig. 3 and 4.   

Response:  As suggested by the reviewer, the symbols in all the figures (Figures 4-8) were increased in size for better visibility.

Finally, the english composition and grammar has been checked by an English professional in the revised manuscript. 

We trust that we have now addressed the major issues raised by the reviewers. My co-authors and I look forward to hearing back from you soon.

Sincerely,

Vaibhav Tiwari Ph.D.

Reviewer 2 Report

In this work, for the first time plant-cell engineered AuNPs have been used in the inhibition of the cell entry regarding two important viruses: HSV-1 and SARS-CoV-2. These nanoparticles have been tested before and after functionalization with quercetin and the different effects on the cell entry  both in case of cells treatment and virus treatment with AuNPs have been assessed. Finally, the same tests have been performed using chemically synthesized commercial amino-functionalized AuNPs, before and after quercetin additional functionalization, for comparison.

In attachment my observations and suggestions.

Author Response

September 19th 2023

Editor
International Journal of Molecular Sciences
Re: Manuscript ID: ijms-2588649

Dear Editor,

We would like to thank you and the reviewers for a prompt review of our manuscript. We are pleased to inform you that we have made the required revisions to the original manuscript by incorporating the required changes as suggested by the reviewers.  Below is the itemized description of changes made in the revised manuscript  following the reviewer’s suggestions.  In addition, we have further provided insight regarding the advantages of plant engineered nanoparticles in viral host adaptation and spread across species. All the changes made to the revised manuscript are highlighted in yellow.

Reviewer 2

Response:  As suggested by the reviewer, we have changed the subheadings in Material & Methods as suggested (highlighted): UV-Visible Spectroscopy, Size and Sigma Potential measurements, Fourier Transform Infra-Red Spectroscopy

Response: Authors agree with Reviewer-2’s comments; there were mistakes in the representation of UV-Vis spectra for pAuNPs and quercetin-coated gold. Reviewer-2 correctly points out that the spectral features lacked correlation with DLS size measurement of the coated gold. We reviewed the spectral measurements of both gold solutions and replaced old representations with the newly constructed Figures: 2 A, B. 

Quercetin excess was removed by centrifugation and only coated nanoparticles were   characterized further. Baseline was adjusted for each spectrum.

Response: As suggested by the reviewer we have now provided quality images                                                           throughout the manuscript. 

Response: Zeta (ζ) potential explanation: As suggested, we explained the negative ζ potential values for both pAuNPs and pAuNPsQ in Results and Discussion (highlighted). We have added a full paragraph on characterization aspects (see highlighted first paragraph of Discussion).

We have reported the zeta potential value of pAuNPs as -23.5mV (Fig. 3A), which further dropped to -28.6 in quercetin coated gold nanoparticles (Fig. 3B). It is important to note the synthesis of pAuNPs occurred within the plant cell (cytoplasm)–without adding any external capping agent–where they bonded with several  functional groups including hydroxyl/alkyl functional groups present in the cytoplasmic environment. The FTIR analysis clearly shows the presence of various functional groups attached to the surface of pAuNPs (Figure 4). Furthermore, we coated those pAuNPs with quercetin that itself carries as many as five –OH groups and some carboxylic group per molecule. Thus, the acidity increased in these anionic nanoparticles, further reducing the ζ potential value to -28.6 mV. We have added references in Discussion section (Biomed Microdevices, 2008, 10:321–328 DOI 10.1007/s10544-007-9139-2).

Quercetin conjugation: We have also shown (Figure 4, panels C, D) that a shift in the hydroxyl peak of pAuNPs occurred from 2977 cm-1 to 2972 cm-1 in quercetin-coated gold (pAuNPsQ), indicating the conjugation of quercetin by hydroxyl-amino linkage. Even after this linkage between amino and hydroxyl groups, several hydroxyl groups of quercetin may have remained free increasing the negative charge of nanoparticle surface.

Additionally, the conjugation of quercetin with particle surface functional groups also ensures good dispersion stability of these nanocomposites, as supported by ζ potential values. We have added some references (ACS Nano. 2009 February 24; 3(2): 386–394. doi:10.1021/nn8005619).

Response: We agree with the reviewer that Transmission Electron Microscopy (TEM) is the best method to determine the physical size of nanoparticles. Thus, in our previous study we focused on the synthesis and characterization of cell-engineered gold nanoparticles confirming the size and shape of each column fraction using TEM-EDS (Bhaskaran et al. 2019; Scientific Reports). In the present study, we used Dynamic Light Scattering (DLS) analysis to specifically determine the hydrodynamic diameter d(H), as the d(H) value gives the more realistic value of particles suspended in liquid while examining their interactions with cell receptors or viral spikes.

Size & Shape: We also learned from our previous study (Bhaskaran et al. 2019; Scientific Reports), that the size and shape of AuNPs depend on the concentration of KAuCl4 added  to alfalfa  cell suspension culture/medium. We thus used the previously standardized condition (KAuCl4: 50 ppm; pH 5.7) of the plant cell culture to obtain a mass of spherical nanoparticles in the size range of 20-22 nm.  Using the DLS method of measurement, in the present study, we again obtained the hydrodynamic diameter of nanoparticles over 21nm (shown in the Results section under Size & Zeta Potential). Commercially available AuNPs of the known specifications (spherical 20 nm) were procured from Nanopartz. Inc, Colorado, USA, for comparisons in the present viral inhibition assay.

Response: We have made the changes in Figure 1 as suggested by the reviewer.

Response: Synthesis panel grouping under Materials & Methods Synthesis figures moved to Material & Methods section as suggested.

Response:  In future, we will aim to investigate the suggested test.

Finally, the english composition and grammar has been checked by an English professional in the revised manuscript. 

We trust that we have now addressed the major issues raised by the reviewers. My co-authors and I look forward to hearing back from you soon.

Sincerely,

Vaibhav Tiwari Ph.D.

Round 2

Reviewer 1 Report

The authors specified all the issues and revised the manuscript. However, there are still some points to be improved:

(1) The current version includes out-of-date references, e.g., ref. 6; ref.13; ref.14; ref.31; ref.33; ref. 35; ref. 36; ref. 43; ref. 46, which can be upgraded. 

(2) The FTIR spectrum captions in panel D Figure 3 can be enlarged or presented as a legend; the cm-1 values and corresponding bond types are poorly visible in this figure version.

Author Response

Review Report 1 (Round 2)

  • As suggested by the reviewer, we have now updated all the references (6, 13, 14, 31, 33, 43, 46) except Ref # 35 and Ref # 36 which are relevant to our studies since coating pAuNPs with quercetin leads to the acidity increased in these anionic nanoparticles further reducing the ζ potential value to -28.6 mV. All the updated/revised references are highlighted in yellow.

  • As suggested by the review, we have now enlarged the FTIR spectrum in both the panel C and panel D (Fig. 3) values and corresponding bond types which are now clearly visible. The newly revised Fig. 3 is now embedded in the text.

Reviewer 2 Report

Dear authors, 

Thank you for your work. 

Author Response

Review Report 2 (Round 2)

  • No comments were made
